# Perceived risk of type 2 diabetes: Using linked genomic, clinical and questionnaire data to understand the potential use of genetic risk tools in British South Asians

Jing Hui Law [1]*, Daniel Stow [1], Sam Hodgson [1], Genes & Health Research Team[2¶], David A. van Heel [2], William G. Newman [3,4], Magda Osman[5], Sarah Finer [1,6]

**1** Wolfson Institute of Population Health, Queen Mary University of London, London, United Kingdom, **2** Blizard Institute, Barts and the London School of Medicine and Dentistry, Queen Mary University of London, London, United Kingdom, **3** Manchester Centre for Genomic Medicine, Manchester University NHS Foundation Trust, Health Innovation Manchester, Manchester, United Kingdom, **4** Division of Evolution, Infection and Genomics, Faculty of Biology, Medicine and Health Sciences, University of Manchester, Manchester, United Kingdom, **5** Judge Business School, University of Cambridge, Cambridge, United Kingdom, **6** Barts Health NHS Trust, London, United Kingdom

¶ Membership of the Genes & Health Research Team is provided in S1 Appendix.
* jing.law@qmul.ac.uk

## Abstract

Despite growing interest surrounding the integration of genetic risk tools such as polygenic risk scores (PRSs) into routine care for early disease identification and management, major questions remain about whether and how these tools are to be implemented at-scale. Many interventions have explored their use in encouraging the adoption of preventative health behaviours—yet existing evidence remains undetermined, limited by the focus on White European populations. The present study used structural equation modelling to explore genetic risk perceptions surrounding type 2 diabetes (T2D) in a sample of British Bangladeshi and British Pakistani volunteers—combining questionnaire data alongside genomic and clinical information to identify the characteristics of individuals who are likely to act on genetic risk information. We conducted this study with volunteers enrolled in Genes & Health—a large-scale ($n > 60,000$) study in the UK recruiting British Bangladeshi and British Pakistani volunteers from community and NHS settings. Eligible participants between the ages of 16 to 59 years were invited to complete a 15-minute questionnaire containing measures of genetic risk perceptions surrounding T2D, as well as intention to adopt health behaviours and that can prevent or delay T2D. Questionnaire responses were then integrated with participants' genomic and clinical data available at Genes & Health to construct a model—characterising their mediating relationships in informing participants' intention. A total of 626 participants responded to the questionnaire (response rate = 17%, 37.70% aged 46 to 59 years, 62.62% female). Being between the ages of 46 to 59 years ($\beta = 0.52$, 95% CI [0.26, 0.79], $p < 0.05$), having greater self-reported perceived control over health ($\beta = 0.41$, 95% CI [0.26, 0.56], $p < 0.05$) and interest in genetic testing ($\beta = 0.62$, 95% CI [0.46, 0.78], $p < 0.05$) all had direct positive effects on participants' intention. Household income showed an indirect

**Data availability statement:** Individual-level data underlying the results presented in this study can be made available to researchers upon application to Genes & Health, following their open access policy described at: https://www.genesandhealth.org/research/scientists-using-genes-health-scientific-research.

**Funding:** This work was made possible by funding from the Wellcome Trust for JHL through the doctoral training programme Health Data in Practice: Human-centred Science (Reference: 218584/Z/19/Z). Genes & Health has recently been core-funded by Wellcome (WT102627, WT210561), the Medical Research Council (UK) (M009017, MR/X009777/1, MR/X009920/1), Higher Education Funding Council for England Catalyst, Barts Charity (845/1796), Health Data Research UK (for London substantive site), and research delivery support from the NHS National Institute for Health Research Clinical Research Network (North Thames). Genes & Health has recently been funded by Alnylam Pharmaceuticals, Genomics PLC; and a Life Sciences Industry Consortium of Astra Zeneca PLC, Bristol-Myers Squibb Company, GlaxoSmithKline Research and Development Limited, Maze Therapeutics Inc, Merck Sharp & Dohme LLC, Novo Nordisk A/S, Pfizer Inc, Takeda Development Centre Americas Inc. The funders had no role in study design, data collection and analysis, decision to publish, or preparation of the manuscript.

**Competing interests:** The authors have declared that no competing interests exist.

effect on intention, mediated by interest in genetic testing, β = 0.24, 95% CI [0.12, 0.37]. Self-identified ethnicity also demonstrated indirect effects on intention via two mediating pathways—both involving participants' T2D PRSs and self-reported family history of T2D (β = 0.03, 95% CI [0.02, 0.05] and β = 0.002, 95% CI [0.001, 0.01]). Our results showed that older age, greater perceived control over health and interest in genetic testing are all predictive of participants' likelihood of adopting preventative heath behaviours in response to genetic risk information about T2D. We also found evidence pointing to the roles that wider socio-demographic, clinical and familial variables can play in informing and mediating genetic risk perceptions. These findings should raise awareness about potential challenges to the equitable delivery and management of genetic risk tools—and strengthen calls for wider family- and system-level approaches that can help address potential health inequalities, as efforts surrounding the large-scale implementation of genomics into existing health systems continue to grow.

## Introduction

Type 2 diabetes (T2D) and its related complications have a disproportionately high prevalence and early onset among individuals of South Asian ancestry [1]. Emerging research has demonstrated that combining genetic risk tools such as polygenic risk scores (PRSs) with QDiabetes—a clinical risk model commonly used in the NHS—can improve the prediction of incident T2D in these populations [2]. Performance is especially enhanced for British Bangladeshi and British Pakistani populations at younger ages and lower body mass index (BMI)—who would otherwise have been considered healthy by QDiabetes alone [2]. Since individuals' genetic liability to T2D is fixed and remains stable from conception, these findings demonstrate the benefit of tools such as PRSs in identifying the degree of predisposition that individuals may have conferred by their genotypes at established T2D-associated genetic loci. This can help inform early disease identification and management—prior to the development and accumulation of clinical and/or lifestyle-related risk factors that conventional clinical risk models rely on [3,4].

At present, PRSs are extensively applied in discovery research—and there is growing interest around their clinical implementation on a population-wide basis [4–7]. Properly translated into clinical settings, improvements in predictive performance can present benefits such as further individualised screening for high-risk individuals and/or earlier referrals onto preventative care [8]. These efforts have been echoed by continuous calls for the wide-scale integration of genomics into routine care in England through the NHS [9–11]. Policies and strategies set out by the Genomic Medicine Service in 2022 describe priority areas such as delivering equitable genetic testing for cancer, rare, inherited and common diseases over the next 5 years—as well as the integration of genomics with other diagnostic and clinical data. As part of these initiatives, there is also a need to better understand the impact of providing individuals with personalised health and risk information using tools such as PRSs—generating evidence that can contribute to decisions on whether and how PRSs are to be implemented at-scale [11].

The clinical utility of PRSs in bringing about downstream population health benefits depend on two key factors: (1) that acting on disease risk can modify an individual's health outcomes; and (2) that the individual informed of their risk may be willing and able to undertake the relevant preventative actions [12]. Many interventions have thus explored the use of genetic risk information in not only shifting individuals' perceptions surrounding a

disease—but also in encouraging the adoption of health behaviours that can prevent or delay disease onset [12–16]. However, current evidence surrounding cardiometabolic diseases is inconclusive—as their multifactorial aetiology and the need for sustained lifestyle changes can pose complex challenges [12,16,17]. Existing research is also limited by the focus on older and healthier White European populations—thus significant gaps remain surrounding how interventions should begin to address wider contextual factors and upstream determinants that may bring about different responses in diverse populations [17]. In particular, whether family experiences with common diseases can correspond to specific motivators for preventative health behaviours has been increasingly acknowledged. This is especially interesting to consider in light of emerging work comparing the interplay of family history information and genome-wide PRSs across 24 common diseases [18]. Family history and PRSs have independent and complementary effects in capturing individuals' risk, highlighting the potential for more comprehensive ways to assess inherited disease risk [18]. There needs to be careful consideration around how these efforts can be translated in practice—as well as how broader influences of risk perceptions and/or health behaviours can be leveraged for the effective communication of genetic risk.

The broad aim of this study was to take a multidisciplinary methodological approach to study genetic risk perceptions surrounding T2D in a sample of British Bangladeshi and British Pakistani volunteers—exploring self-reported questionnaire data in the context of wider contextual factors, including participants' actual genomic and clinical information. We undertook our study with volunteers enrolled in Genes & Health—a large-scale biobank in the UK which has recruited over 60,000 British Bangladeshi and British Pakistani participants from community and NHS settings [2,19]. Combining the rich data resource in Genes & Health with a large-scale questionnaire on genetic risk perceptions with 626 volunteers, we applied structural equation modelling (SEM) as our main analysis. Specific aims were to (1) characterise the mediating relationships between various socio-demographic, genomic, clinical and questionnaire variables; and (2) identify the characteristics of individuals who are likely to act on genetic risk information about T2D.

## Materials and methods

### Design

Our integrated study design and analysis brought together multiple data sources in Genes & Health—investigating questionnaire-derived genetic risk perceptions alongside the genomic and clinical datasets available. Volunteers who were already enrolled in the biobank were invited to complete a cross-sectional, online-based questionnaire on genetic risk perceptions surrounding T2D via the online questionnaire platform REDCap. Questionnaire responses were then linked to genomic and clinical data via participants' pseudonymised NHS numbers.

### Ethical statement

Genes & Health operates under ethical approval from the London South East National Research Ethics Committee (REC) and Health Research Authority (HRA), with Queen Mary University of London as Sponsor [19]. Details about the cohort have been described elsewhere [2,7,19]—and an overview of its recruitment process is included in S2 Appendix. In brief, British Bangladeshi and British Pakistani volunteers aged 16 years and above have donated saliva samples for DNA extraction and genetic tests, provided consent for researchers to access their electronic health records (EHRs), as well as consent to be recontacted (up to four times per year) for recall studies via separate ethics applications. The present study is linked to its original REC/HRA approvals (14/LO/1240). An application was first made to the Genes & Health

Executive for internal review—and then submitted as an ethics amendment to REC/HRA. Approval was obtained on the 15th of August 2022. All participants had to provide electronic written consent by checking the consent form on REDCap before responding to our questionnaire (S3 Appendix). For participants under the age of 18, consent from parents or guardians was not required, as the low risk nature of this study is covered by existing approvals in Genes & Health for the recall of all its adult volunteers aged 16 years and above (S2 Appendix).

## Participants

To be eligible for our questionnaire, participants had to:

1. Be between the ages of 16 to 59 years;

2. Have no previous diagnosis of T1D or T2D in their primary care records; and

3. Have an email address and/or phone number registered with Genes & Health.

Volunteers meeting the above inclusion criteria were identified using their linked and pseudonymised demographic and health data stored in the Genes & Health Trusted Research Environment (TRE), as of the July 2022 data release. Details about our eligibility screening process are also presented in S2 Appendix.

## Measures

**Questionnaire data.** An overview of the questionnaire is presented in Table 1—with details of specific items and scoring methods described in S3 Appendix. These questions encompass measures of genetic risk perceptions such as individuals' perceived risk of T2D, interest in genetic testing, as well as their perceived control over their own health. It also includes familial variables such as whether individuals have any known family members with a history of T2D, as well as questions about health behaviours that are present in their family environment. Most of these questions were adapted from established and validated measures of genetic risk perceptions available in the literature (newly developed measures will be specifically indicated in S3 Appendix)—and then further refined and optimised via Patient and Public Involvement (PPI). Prior to recruitment, we conducted workshops and one-to-one PPI sessions with volunteers of Bangladeshi and Pakistani descent in the UK (including those not involved in Genes & Health) to test and develop the questionnaire iteratively. These sessions revolved predominantly around checking understandability and acceptability of the

**Table 1. Overview of questionnaire items.**

| Category | Items |
|---|---|
| Genetic risk perceptions | Knowledge of the genetic basis of T2D |
| | Perceived risk for T2D |
| | Interest in genetic testing |
| | Perceived benefits of genetic testing |
| | Perceived control |
| Familial variables | Known family members and/or close social contacts with a history of T2D |
| | Family health behaviours |
| Outcome variables | (Primary intention outcome) Intention to adopt health behaviours that can prevent or delay T2D, if a genetic test shows above-average risk for the condition |
| | (Secondary intention outcome) Interest in receiving an email about further online resources on health behaviours that can prevent or delay T2D |

questionnaire within our target population—as well as ensuring that questions are culturally sensitive and relevant to participants' understanding of T2D. Where required, bilingual staff at Genes & Health were involved in these PPI sessions to aid with translations between Bengali/ Urdu and English. The questionnaire has also been extensively reviewed by members of the Genes & Health research team, as well as the Genes & Health Community Advisory Group— to help identify any potential difficulties or sensitivities with the questionnaire items, as well as to oversee the feasibility of the study.

**Genomic data.** Genotyping in Genes & Health was performed on Illumina Infinium Global Screening Array v3 with additional multi-disease variants. Variants with call rates < 0.99 and/or minor allele frequencies < 1% were excluded, as were single nucleotide polymorphisms with imputation quality scores < 0.3. We excluded individuals unlikely to have Pakistani or Bangladeshi ancestry on the basis of principal component 1 lying 3 or more standard deviations (SDs) from the self-reported mean. Imputation was performed using the TopMED-r2 panel. We estimated participants' genetic risk for T2D using externally-derived scores published in the PGSCatalog as of April 2023 [20]. We selected the best-performing score, assessed by beta estimated from multivariate logistic regression models adjusted for age, sex, ancestry, and the first 20 genetic principal components; this outperformed scores developed within Genes & Health, without risk of overfitting. This was a European-ancestry score comprising 6,437,380 variants derived in 898,130 individuals [21]. The score was scaled to a normal distribution with a mean and median of 0 and SD of 1 in the broader Genes & Health population. The sample mean is reported alongside descriptive statistics in Table 2 in the results section. We used this score for subsequent linkage and analysis in the present study—but they were not directly fed back to participants (i.e., participants were not made aware of their T2D PRSs).

**Clinical data.** Clinical data extracted from participants' EHRs included BMI and the presence of comorbidities. BMI was obtained from their primary or secondary care records up to December 2022—to capture values closest to the point of questionnaire recruitment. Comorbidities were extracted using SNOMED and ICD-10 codes—and with codelists generated as part of the NIHR AI MULTIPLY consortium [22]. A list of 22 physical and psychological health conditions were selected—guided by the NHS Quality and Outcomes Framework clinical and public health indicators for 2023/24 in England, based on evidence of health conditions that are likely to benefit from improved primary care [23]. This encompassed conditions such as asthma, atrial fibrillation and hypertension (full list in S4 Appendix).

**Socio-demographic data.** Questions about participants' highest level of education, annual household income and the number of people living in their household were included towards the end of our questionnaire (S3 Appendix). These were adapted from UK Biobank material (available on the UK Biobank online resource centre). Additional demographic information— including self-identified sex and ethnicity—was extracted from participants' Genes & Health Stage 1 recruitment questionnaires.

## Procedure

Sample size calculations for the questionnaire were based on the original paper from which the primary intention outcome was extracted [24]. Details are presented in S2 Appendix, alongside further information about our recruitment strategy. A stratified random sampling approach was taken to ensure balanced representation across age and sex (S2 Appendix). We first identified a total of 5,000 eligible volunteers. After filtering out those who have recently been recalled for other studies in Genes & Health, 4,955 questionnaire invitations were scheduled (Fig 1). These were sent out via email and/or text messages, with questionnaire

**Table 2. Participant characteristics.**

| Participant characteristics (N = 626) | *n* (%)/ Mean (SD) |
|---|---|
| Age group | |
| 16 to 25 years | 136 (21.73) |
| 26 to 35 years | 123 (19.65) |
| 36 to 45 years | 131 (20.93) |
| 46 to 59 years [a] | 236 (37.70) |
| Sex | |
| Male | 234 (37.38) |
| Female | 392 (62.62) |
| Ethnicity | |
| Bangladeshi [a] | 338 (53.99) |
| Pakistani | 281 (44.89) |
| Other | 7 (1.12) |
| Diabetes history | |
| Gestational diabetes | 39 (6.23) |
| Pre-diabetes | 63 (10.06) |
| No diabetes history | 524 (83.71) |
| Known family members and/or close social contacts with a history of T2D | |
| First degree family members | 357 (57.03) |
| Other close social contacts | 269 (42.97) |
| Number of people living in household | 4.52 (1.69) |
| Household income | |
| Less than £18,000 [a] | 168 (26.84) |
| £18,000 to £30,999 | 114 (18.21) |
| £31,000 to £51,999 | 113 (18.05) |
| £52,000 to £100,000 | 78 (12.46) |
| Greater than £100,000 | 18 (2.88) |
| Do not know/prefer not to say | 135 (21.57) |
| Education level | |
| College or University degree [a] | 340 (54.31) |
| A levels/AS levels or equivalent | 113 (18.05) |
| O levels/GCSEs or equivalent | 57 (9.11) |
| CSEs or equivalent | 12 (1.92) |
| NVQ or HND or HNC or equivalent | 24 (3.83) |
| Other professional qualifications (e.g., nursing, teaching) | 13 (2.08) |
| None of the above/prefer not to say | 67 (10.70) |
| T2D PRSs (missing *n* = 117; 18.69%) | − 0.13 (1.00) |
| BMI (missing *n* = 220; 35.14%) | 25.92 (4.92) |
| Number of comorbidities | 1.42 (1.36) |
| Genetic risk perceptions | |
| Knowledge of the genetic basis of T2D | 19.9 (3.87) |
| Perceived risk for T2D | 2.83 (0.90) |
| Interest in genetic testing | 8.05 (1.12) |
| Perceived benefits of genetic testing | 18.1 (2.65) |
| Perceived control | 11.9 (1.66) |
| Family health behaviours | 27.2 (4.81) |
| Primary intention outcome | 10.4 (1.58) |

*(Continued)*

**Table 2.** (Continued)

| Participant characteristics (N = 626) | *n* (%)/ Mean (SD) |
|---|---|
| Secondary intention outcome | |
| Interested in receiving further online resources T2D prevention | 431 (68.85) |
| Not interested in receiving further online resources T2D prevention | 195 (31.15) |

[a]Indicator group for multilevel variables included in SEM analysis.

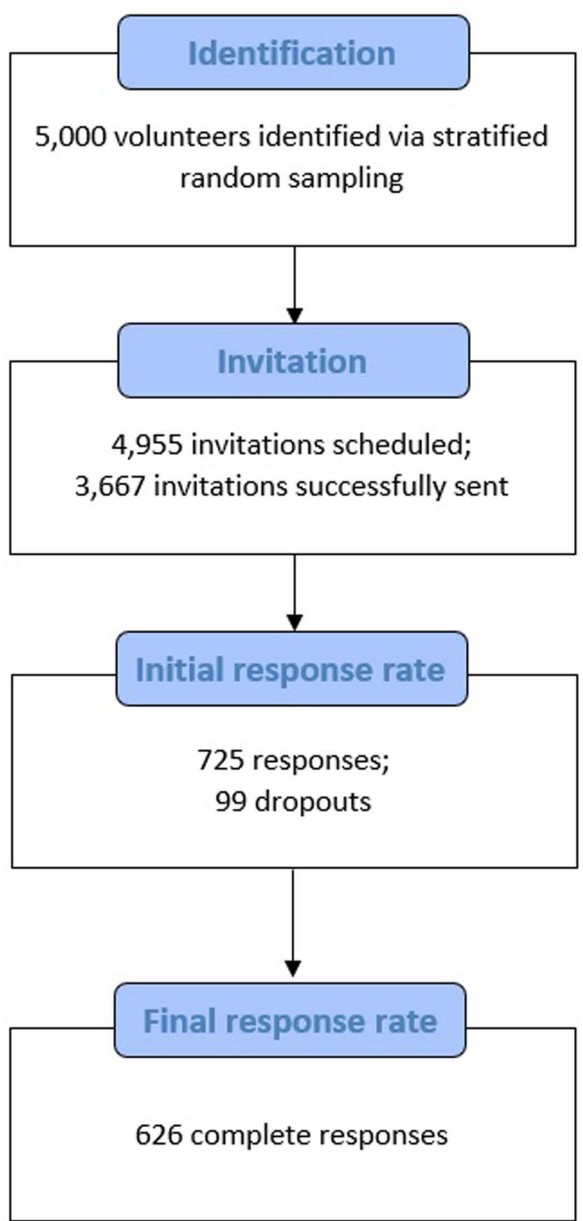

**Fig 1. Flowchart illustrating participant recruitment.**

links uniquely integrated—such that participants who received both email and text message invitations could only access and complete the questionnaire once.

An overall bounce rate of approximately 26% was observed across the invitations. Of the 3,667 invitations successfully sent out, 725 individuals started the questionnaire and 99 respondents dropped out. A total of 626 complete responses were collected, giving a final response rate of 17% (626/3,667). This figure is comparable to recruitment rates that have been reported for online questionnaire studies in other large genomic cohorts such as FinnGen (18.6%) [25]. Additionally, a comparison of the relevant genomic and clinical data between responders and non-responders in our sample suggest that they do not differ significantly—at least in terms of their T2D PRSs, BMI and number of comorbidities (S5 Appendix).

On the REDCap landing page for the questionnaire, participants were first presented with a participant information sheet and consent form (S3 Appendix). Those who provided electronic written consent by checking the form were then directed to answer further screening questions about age and diabetes history to fully ensure eligibility. Only participants who met these inclusion criteria could then proceed with the questionnaire. The questionnaire took approximately 15 minutes to complete and participants were reimbursed with a £15 voucher. Recruitment began on the 18th of August 2022 and lasted until the 4th of January 2023. Upon closing the questionnaire, participants' responses were securely exported from REDCap as an Excel data file—then uploaded back onto the TRE on the 4th of April 2023 for data linkage and subsequent analysis. Participants were not identifiable during or after these data collection processes, as all questionnaire data and linked information in the TRE were pseudonymised.

## Data analysis

Descriptive analysis was first conducted to explore participant characteristics, as well as their genomic and clinical data obtained via linkage in Genes & Health. For the main analysis, SEM was performed using R packages "lavaan" [26] and "semTools" [27] to define and test a theoretical model incorporating all socio-demographic, genomic, clinical and questionnaire variables described above. SEM is a statistical procedure widely used to formalise and explore structural relationships between networks of variables and abstract constructs that cannot be directly measured or observed. Given the nature of constructs such as perceived control and intention involved in the current study, SEM allowed the opportunity to define and test mechanisms between these hypothetical constructs—estimating any direct or indirect relationships, whilst accounting for any potential measurement errors. Our hypothesised model is shown in Fig 2 below. Further details around the specified relationships between these variables are provided in the following subsections.

**Confirmatory factor analysis.** All multiple-item measures in the questionnaire were originally included as latent variables in the model—including participants' knowledge of the genetic basis of T2D, interest in genetic testing, perceived control, perceived benefits of genetic testing, family health behaviours and the primary intention outcome. Confirmatory factor analysis was conducted to define their measurement model and assess the validity of these latent variables. Cronbach's alpha coefficients were also obtained to evaluate measurement reliability. The final measurement model is shown in Fig 3, with further details presented in S6 Appendix. Interest in genetic testing was reduced into a one-item construct. Perceived control and perceived benefits of genetic testing were combined into a single construct, due to some overlap in the dimensionality of the two sets of questions. Additionally, only three items were retained in the final measurement for the family health behaviours construct. This model showed a good fit, $\chi^2(95) = 166.46$, $p < 0.05$, CFI = 0.98, TLI

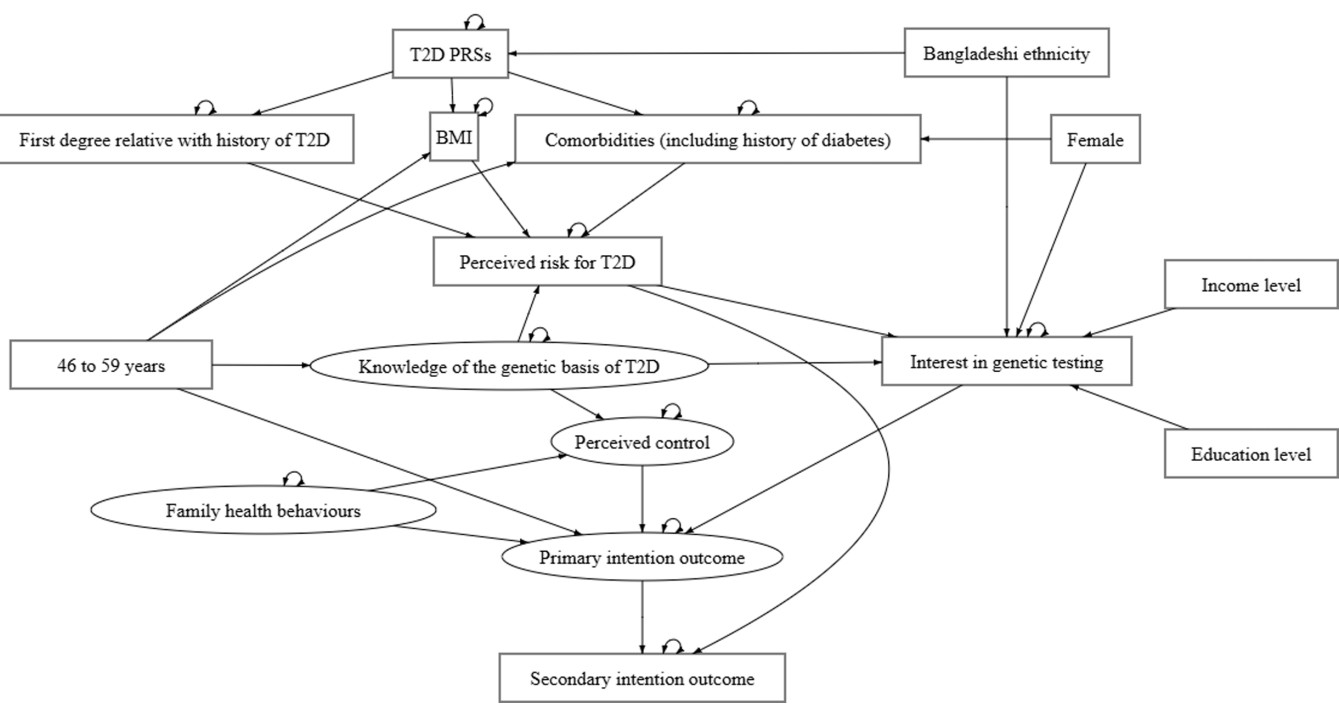

**Fig 2. Hypothesised model for SEM analysis.**

= 0.98 and RMSEA = 0.04, 90% CI [0.03, 0.04] (Fig 3). Curved double-headed arrows in the figure represent variance or covariance.

**Structural equation modelling.** The main exogenous variable included in our hypothesised model as shown in Fig 2 was a binary variable for age—according to the age group with the largest *n* in our sample—to aid interpretation (46 to 59 years = 1; all other age groups = 0). Measurements of genetic risk perceptions, as well as the primary and secondary intention outcomes were included as endogenous variables. The relationships between these variables were defined based on prominent theories and frameworks that have been applied from the field of health psychology into the literature on genetic risk perceptions (e.g., the Theory of Planned Behaviour; the Health Belief Model). However, acknowledging that existing theories in the literature may not be sufficient to allow for a full understanding of health-related perceptions and beliefs in British Bangladeshi and British Pakistani populations, we have adapted a model specifying more complex relationships between these genetic risk perceptions—and expanded on this by further integrating external variables with plausible influence on individuals' perceptions and behaviours. Thus, whilst building on existing theories, the goal of our analysis was to chart out a more complex picture of genetic risk perceptions in this underrepresented sample—evaluating and appraising the fit of our model against these theoretical propositions.

These external variables included participants' self-identified ethnicity, sex, as well as data extracted from their questionnaire responses on family history of T2D and family health behaviours. Being of Bangladeshi ethnicity and self-identifying as female were coded for as the indicator variables for ethnicity and sex, respectively—as these were the categories with the larger *n* in the sample. Socio-demographic variables such as annual household income and highest level of education were also coded for categories with the largest *n*—"Less than £18,000" and "College or University degree", respectively. Number of comorbidities in the

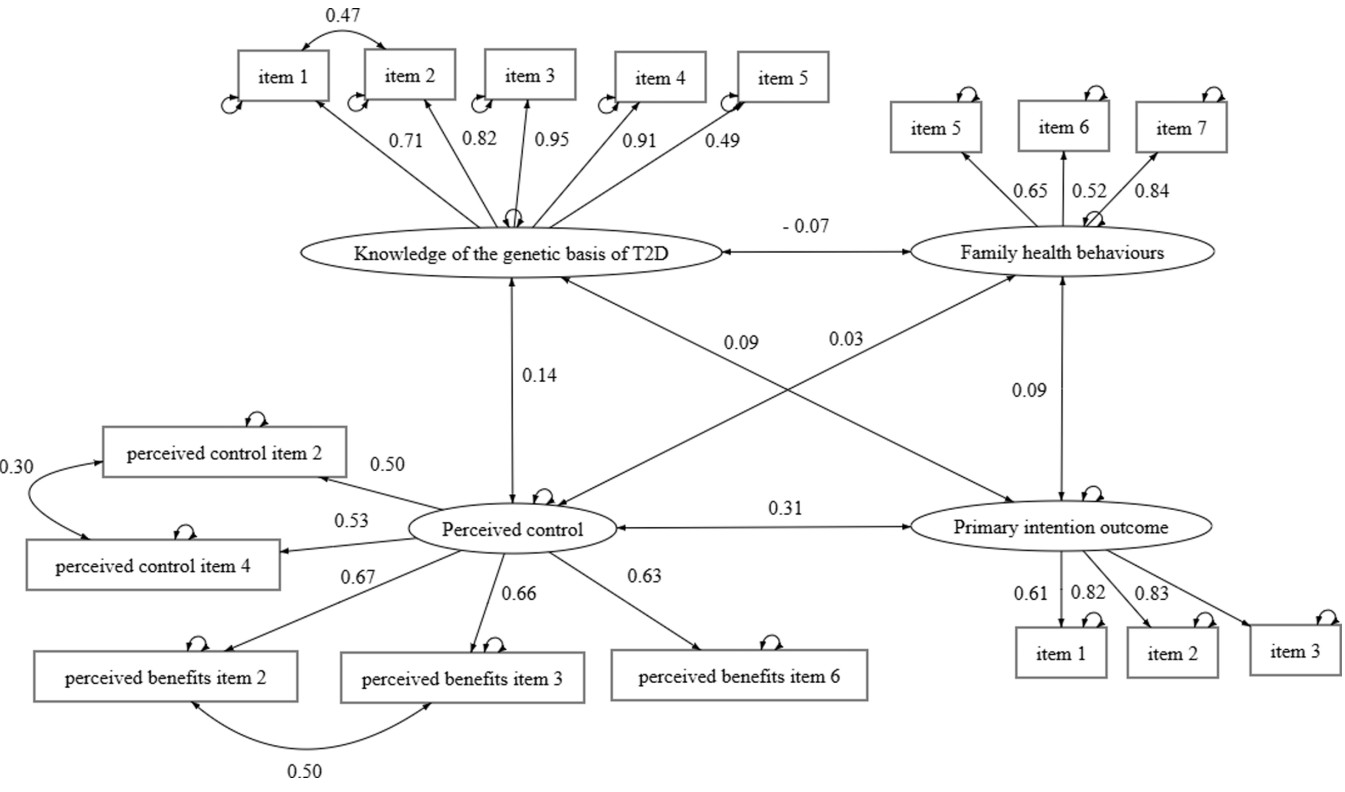

**Fig 3. Final measurement model for latent variables.**

model was adapted to account for participants' self-reported history of gestational diabetes and/or pre-diabetes from the questionnaire.

There was missing data in participants' T2D PRSs ($n$ = 117; 18.69%) and BMI ($n$ = 220; 35.14%) in our sample. We examined the relationship between this missingness and other observed variables included in our model to determine the pattern of missingness. Analysis indicated that data for both T2D PRSs and BMI were missing at random—i.e., the probability of missing values for both variables were significantly associated with other key variables included in our model, such as age group and ethnicity ($p$ <.05). Multiple imputation by predictive mean matching (PMM) was thus applied alongside SEM, using R package "mice" [28] to create and analyse 30 imputed datasets—estimating missing values based on all other variables included in our model (e.g., age, sex and family history of T2D). PMM as an approach is largely considered to be robust to model misspecification [29]—and can perform well with up to 50% missingness [30]. Still, the possibility of bias caused by high rates of missingness in our data (particularly with BMI) should not be overlooked. Details around imputation diagnostics—demonstrating that convergence has been achieved in our model—are included in S7 Appendix. The inclusion of both T2D PRSs and family history of T2D in our model may also introduce confounding bias. However, in light of emerging evidence on their independent and complementary effects on T2D susceptibility [18], it was in our interest to examine the potential mediating relationships between T2D PRSs and family history of T2D. Due to the number of ordinal variables involved in the latent constructs defined in our analysis, diagonally weighted least squares estimation was used to fit our hypothesised model as shown in Fig 2. All analysis was conducted using R version 4.2.1 [31].

## Results

Participant characteristics are presented below, alongside descriptive statistics for the genomic and clinical data obtained via linkage in Genes & Health (Table 2). Participants scores on the measures of genetic risk perceptions, as well as on the primary intention outcome, are presented in their original, numerical form at this stage (details about scoring methods in S3 Appendix). Where applicable, the indicator group for the multilevel variables included in our SEM analysis have been specified accordingly.

Our hypothesised model demonstrated good fit across three of the imputed datasets, $\chi^2(323) = 622.90$, $p < 0.05$, CFI = 0.97, TLI = 0.98 and RMSEA = 0.04, 90% CI [0.03, 0.04]. All pooled estimates are presented in Fig 4 below. Significant effects are denoted with solid lines ($p < 0.05$) and non-significant effects are denoted with dashed lines.

### Direct effects

Being in the oldest age group ($\beta = 0.52$, 95% CI [0.26, 0.79], $p < 0.05$), having greater perceived control ($\beta = 0.41$, 95% CI [0.26, 0.56], $p < 0.05$) and greater interest in genetic testing ($\beta = 0.62$, 95% CI [0.46, 0.78], $p < 0.05$) all had positive direct effects on the primary intention outcome—i.e., participants' self-reported intention to adopt health behaviours that can prevent or delay T2D, if a genetic test shows that they are at above-average risk for T2D. This, in turn, showed a positive direct effect ($\beta = 0.19$, 95% CI [0.10, 0.28], $p < 0.05$) on participants' interest in receiving an email about further online resources on health behaviours that can prevent or delay T2D—a measure defined as the secondary intention outcome in this study.

In terms of other significant relationships between the genetic risk perceptions measured in the questionnaire, perceived risk had a positive direct effect on the secondary intention outcome ($\beta = 0.25$, 95% CI [0.15, 0.34], $p < 0.05$). Additionally, participants' perceived control

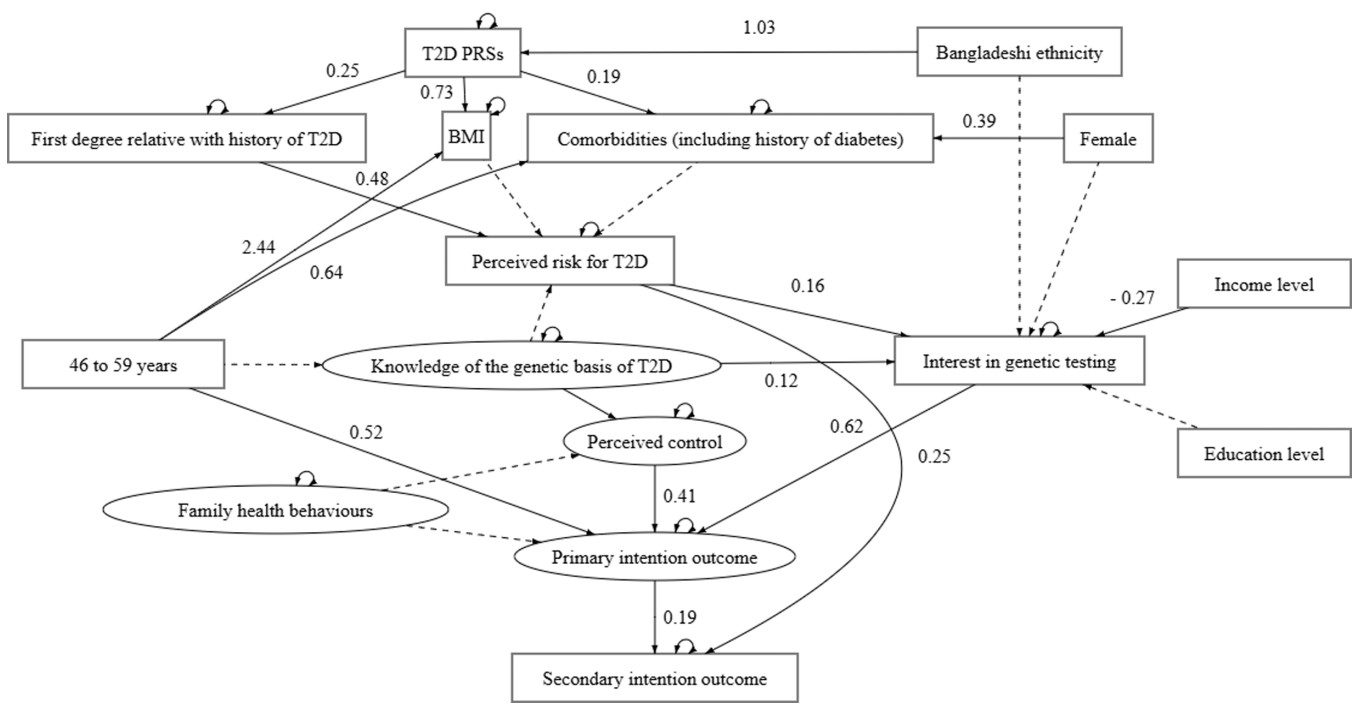

**Fig 4. Results of SEM analysis.**

and interest in genetic testing were both predicted by their self-reported knowledge of the genetic basis of T2D—β = 0.16, 95% CI [0.06, 0.25], $p < 0.05$ and β = 0.12, 95% CI [0.02, 0.21], $p < 0.05$, respectively. Interest in genetic testing was significantly predicted by perceived risk (β = 0.16, 95% CI [0.06, 0.25], $p < 0.05$) and income level (β = − 0.27, 95% CI [− 0.53, − 0.01], $p < 0.05$). Perceived risk, in turn, was predicted by having first degree family member(s) with a history of T2D (β = 0.48, 95% CI [0.38, 0.57], $p < 0.05$).

Hypothesised relationships between the other genomic and clinical variables included in our model indicated that participants' T2D PRSs had direct effects on BMI (β = 0.73, 95% CI [0.26, 1.20], $p < 0.05$), comorbidities (β = 0.19, 95% CI [0.07, 0.31], $p < 0.05$) and having first degree family member(s) with a history of T2D (β = 0.25, 95% CI [0.15, 0.35], $p < 0.05$). Furthermore, being of self-identified Bangladeshi ethnicity was associated with having higher T2D PRSs, in comparison to being of Pakistani descent (β = 1.03, 95% CI [0.86, 1.19], $p < 0.05$). Comorbidities were predicted by sex (β = 0.33, 95% CI [0.07, 0.58], $p < 0.05$). Additionally, being in the oldest age group had direct effects on both BMI (β = 2.44, 95% CI [1.52, 3.36], $p < 0.05$) and comorbidities (β = 0.64, 95% CI [0.40, 0.89], $p < 0.05$).

## Mediation effects

In examining mediational pathways, Monte Carlo simulations were generated with 1,000 samples to yield robust 95% confidence intervals for the indirect effects, based on the estimated model parameters. The mediating pathways tested are presented in S8 Appendix. Being in the oldest age group had a significant positive indirect effect on the secondary intention outcome, through its influence on the primary intention outcome, β = 0.10, 95% CI [0.04, 0.17] (S7 Fig in S8 Appendix). Income level also had an indirect effect on the outcome measures, mediated by participants' self-reported interest in genetic testing, β = 0.24, 95% CI [0.12, 0.37] (S8 Fig in S8 Appendix). Additionally, participants' self-identified Bangladeshi ethnicity had positive indirect effects on the questionnaire outcomes via two mediating pathways. Firstly, its effect on the secondary intention outcome was mediated by T2D PRSs, having first degree family member(s) with a history of T2D and perceived risk, β = 0.03, 95% CI [0.02, 0.05] (S9 Fig in S8 Appendix). Secondly, this relationship was also mediated by T2D PRSs, having first degree family member(s) with a history of T2D, perceived risk, interest in genetic testing and the primary intention outcome, β = 0.002, 95% CI [0.001, 0.01] (S10 Fig in S8 Appendix).

## Discussion

This study aimed to take a multidisciplinary methodological approach to investigate T2D genetic risk perceptions in British Bangladeshi and British Pakistani volunteers enrolled in Genes & Health—exploring questionnaire data in the context of participants' actual genomic and clinical information. Combining the rich data resource in Genes & Health with a large-scale questionnaire with 626 volunteers, SEM was performed as the main analysis to define and test a theoretical model—incorporating various socio-demographic, genomic and clinical variables to characterise their mediating relationships alongside participants' T2D genetic risk perceptions—and using this to identify the characteristics of individuals who are likely to act on genetic risk information about T2D.

Our results suggest that participants in the oldest age group (46 to 59 years) tended to report greater intention to adopt health behaviours that can prevent or delay T2D, if a genetic test were to show that they are at above-average risk for the condition. This ultimately fed into their interest in receiving an email about further online resources on T2D prevention—an outcome which was taken as a secondary measure of intention in the questionnaire. This finding is consistent with previous work suggesting that older populations usually demonstrate

greater interest in seeking out disease risk information via genetic testing, compared to younger people [14,15]. The saliency and relevance of health risks are often stronger in older adulthood—as this usually represents the stage of life where health concerns may first be emerging, yet there may still be time for behavioural changes to have a positive impact on health states [15]. Some studies have further reported that, compared to younger individuals, older adults usually demonstrate significantly stronger consistency between their self-reported intention—and the actual adoption of health behaviours at follow-up [32]. Such findings have been attributed to older adults having more established routines and habits—which may contribute to the regularity of their lifestyles and behaviours. Younger adults, on the other hand, are more likely to experience significant life adjustments (e.g., changes to living situations; forming new relationships)—which can explain the lack of alignment between their intention and behaviours [32]. It is worth noting, however, that when asked about their preferred age for genetic testing in our questionnaire, most participants (34.5%) still indicated that they would like to find out at younger ages—between 16 to 29—if they were genetically at risk of T2D. This suggests a certain level of discordance between the age at which individuals would want to find out about their risk, versus the age at which they would actually be ready—or willing—to implement preventative lifestyle changes.

Findings from our model also shed light on some of the roles that wider socio-demographic, clinical and familial variables can play in informing and mediating genetic risk perceptions surrounding T2D. For example, there were mediating effects between participants' self-reported household income and interest in genetic testing in predicting outcomes on the questionnaire. Although overall effects were positive, that participants from lower income households in our sample reported less interest in genetic testing (even if these genetic tests were being offered for free) largely echo the gaps in health-related perceptions and behaviours that may exist between different socio-demographic backgrounds. This phenomenon should raise awareness about potential challenges to the equitable delivery and management of genetic risk tools. Whether due to a lack of awareness or understanding around the uses or purposes of genetic testing—or even perceived challenges or barriers to access health services—such differences point to the influence that characteristics linked to deprivation can have in driving health inequities. It also highlights the need for any efforts to integrate large-scale genetic testing on a population level to account for the social determinants of health—in order to fully maximise the utility and benefits of genetic risk tools. If PRSs are to be implemented at-scale, strategies will need to also target system-level factors—tackling the pathways and mechanisms driving potential health inequalities—so that individuals are not placed at further disadvantage.

Additionally, relationships between participants' T2D PRSs and perceived risk for T2D in our study sample were predominantly mediated by having first degree family member(s) with a history of T2D, even when compared to clinical factors such as BMI or comorbidities. These results suggest that the relationship between participants' genetic risk of T2D and their heightened sense of perceived risk is predominantly exerted through observing other family members with T2D—even after accounting for other clinical variables. It highlights that previous experiences with T2D in the family context can impact on how individuals think about their own risk—with further downstream effects on how they might readily react to genetic risk information. Further work exploring how this can be leveraged in the communication of genetic risk is warranted, as there may be opportunities for interventions to tap into unique family experiences as drivers in encouraging preventative health behaviours. Other findings from our model showed that T2D PRSs are, in turn, predicted by being of self-identified Bangladeshi ethnicity. Additionally, female participants had more comorbidities recorded in their EHRs.

Whilst discussions around the wide-scale integration of genomics into routine care in the NHS in England have been gaining traction [9–11], our results lend support to the idea that the provision of genetic risk information should be combined with other forms of support to achieve goals of motivating preventative health behaviours more widely. There are already pilot trials being implemented to explore the integration of PRSs for cardiovascular disease into NHS Health Checks—the national programme offering free health checks every 5 years to adults between the ages of 40 to 74 in primary care [33,34]. However, the operational and logistical impact of incorporating genomic information into these settings will require careful assessment and planning across various services and resources to ensure equitable use. There may be potential for educational interventions to be integrated alongside PRS delivery—to ensure that varying levels of understanding and/or interest in individuals of diverse socio-demographic backgrounds are addressed. Interventions can also be supplemented with further system-level services to facilitate the translation from risk awareness into actual preventative action across diverse groups [17]. Such efforts may incorporate elements such as environmental restructuring or social planning to address wider issues such as the availability of affordable healthy food, or to improve individuals' access to preventative support. This is especially important for individuals who may not necessarily have adequate resources and opportunities to implement lifestyle changes, despite being at high risk. Additionally, given interest in genetic testing was measured in this study with participants' assumption that these genetic tests would be freely available—ensuring that the costs of any such services remain accessible to the public will be an important factor to address. In this respect, significant developments and funding will be required for existing health systems—whether in the NHS or across global contexts—to ensure that the necessary resources and infrastructures can be implemented to support the future application of PRSs for common diseases such as T2D.

Furthermore, building on the specific findings surrounding familial variables in our model, the potential role of family-based cascade testing may be an approach to consider for risk assessment. Unaffected family member(s) of patients who are already diagnosed with T2D likely already perceive themselves to be at high risk—thus baseline readiness to engage with preventative health behaviours may be higher than the general population—and offering genetic risk information about the condition may present unique benefits. However, how these relationships might play out in younger individuals at risk of T2D will need to be considered. Given PRSs often demonstrate the strongest clinical utility in younger populations [3,4], current findings around the discordance between the age at which individuals would want to find out about their genetic risk versus the age at which they would actually be willing to engage in preventative health behaviours are worth further exploration. It may be that, for younger populations to fully benefit, greater efforts combining system-level and environmental support services—integrating family, social and other resources—to fully engage preventative health behaviours will be needed to supplement the implementation of PRSs.

There were some limitations in this study—perhaps predominantly with regards to the representativeness of our questionnaire sample. It must be noted that respondents were recruited from a consented cohort, already involved in a large-scale genomics and health study. Although Genes & Health can broadly be considered representative of its background population [19], female participants are slightly overrepresented in this sample ($n = 392$; 62.62%)—and self-reported levels of genetic risk perceptions found here may not necessarily reflect the views of underlying British Bangladeshi and British Pakistani populations in the UK. Future work should aim to explore these issues more widely. The possibility of missing data in participants' T2D PRSs and BMI introducing bias should also be considered. Imputation diagnostics included in S7 Appendix demonstrate that convergence has been achieved, but future work involving a larger sample size where complete-case analysis is viable may help

generate stronger arguments about the mediating relationships found in this study. The inclusion of both T2D PRSs and family history of T2D in our SEM analysis may have introduced confounding bias. However, as mentioned in our methods section, it was in our interest to examine the potential mediating relationships between T2D PRSs and family history of T2D.

Additionally, the original psychometric properties of some questionnaire measures that we have taken from the literature for this study have been affected following PPI procedures. However, these have been modified and refined according to item statistics (S6 Appendix)—with inadequate items that were limited in terms of validity or reliability removed before inclusion in the final model. On a similar note, the educational and income categories previously defined in UK Biobank might not necessarily reflect the same groups in Genes & Health. Descriptive statistics showed that whilst most participants in our sample reported an annual household income of less than £18,000, over half of the sample also reported being at least university- or college-educated. There may be other factors to consider here, such as participants' immigration status, the countries where their educational qualifications were obtained—and whether such qualifications are equivalent to traditional definitions of being college- or university-educated in the UK. In future work, it may be that even seemingly objective socio-demographic measures also need to undergo a process of tailoring and refining. This can help ensure that variables such as educational and income categories can be accurately captured in diverse samples.

Nevertheless, this study has been able to provide some novel and unique insights into the perspectives surrounding genetic risk for T2D in an underrepresented population so disproportionately affected by the condition. It has also leveraged the rich genomic and clinical data available at Genes & Health to begin charting out the complexity of relationships underpinning these genetic risk perceptions. Taken together, our results point to the important roles that upstream determinants and contextual factors such as family history and household income can play in leveraging the use of genetic testing for T2D in British Bangladeshi and British Pakistani populations. As efforts surrounding the large-scale implementation of genomics into existing health systems continue to grow, future work should explore ways to integrate wider family- and system-level approaches that can help address potential health inequalities. In the present study, we also found that there was a discordance between the age at which individuals would want to learn about their genetic risk of T2D—versus the age at which they would be ready or willing to implement preventative lifestyle changes. It will be important to consider how the integration of genomics into heath systems can be further tailored to meet the needs of younger populations.

## Supporting information

**S1 Appendix. Membership of the Genes & Health Research Team.**
(DOCX)

**S2 Appendix. Recruitment process in Genes & Health and the present study.**
(DOCX)

**S3 Appendix. Questionnaire items.**
(DOCX)

**S4 Appendix. Comorbidities included in analysis.**
(DOCX)

**S5 Appendix. Comparison of genomic and clinical data between responders and non-responders.**
(DOCX)

**S6 Appendix. Results from confirmatory factor analysis.**
(DOCX)

**S7 Appendix. Imputation diagnostics.**
(DOCX)

**S8 Appendix. Mediating pathways tested.**
(DOCX)

## Acknowledgments

We thank Social Action for Health, Centre of The Cell, members of our Community Advisory Group, and staff who have recruited and collected data from volunteers. We thank the NIHR National Biosample Centre (UK Biocentre), the Social Genetic & Developmental Psychiatry Centre (King's College London), Wellcome Sanger Institute, and Broad Institute for sample processing, genotyping, sequencing and variant annotation.

We thank: Barts Health NHS Trust, NHS Clinical Commissioning Groups (City and Hackney, Waltham Forest, Tower Hamlets, Newham, Redbridge, Havering, Barking and Dagenham), East London NHS Foundation Trust, Bradford Teaching Hospitals NHS Foundation Trust, Public Health England (especially David Wyllie), Discovery Data Service/Endeavour Health Charitable Trust (especially David Stables), Voror Health Technologies Ltd (especially Sophie Don), NHS England (for what was NHS Digital)—for GDPR-compliant data sharing backed by individual written informed consent.

Most of all we thank all of the volunteers participating in Genes & Health.

## Author contributions

**Conceptualization:** Jing Hui Law, Magda Osman, Sarah Finer.

**Data curation:** Jing Hui Law, Daniel Stow, Sam Hodgson.

**Formal analysis:** Jing Hui Law.

**Funding acquisition:** Jing Hui Law.

**Methodology:** Jing Hui Law, Daniel Stow, Sam Hodgson, Magda Osman, Sarah Finer.

**Project administration:** Jing Hui Law.

**Supervision:** David A van Heel, Magda Osman, Sarah Finer.

**Visualization:** Jing Hui Law.

**Writing – original draft:** Jing Hui Law.

**Writing – review & editing:** Jing Hui Law, Daniel Stow, Sam Hodgson, David A van Heel, William G Newman, Magda Osman, Sarah Finer.

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
