## [Decision Letter · Decision Letter 0]

6 Aug 2024

PGPH-D-24-00893

Perceived risk of type 2 diabetes: Using linked genomic, clinical and questionnaire data to understand the potential use of genetic risk tools in British South Asians

Dear Dr. Law,

Thank you for submitting your manuscript to PLOS Global Public Health. Firstly, we would like to apologize for the delay in processing your manuscript. It has been exceptionally difficult to secure reviewers to evaluate your study. We have now received one completed review, which is available below. The reviewer has raised significant scientific concerns about the study that need to be addressed in a revision.

Please note that we have only been able to secure a single reviewer to assess your manuscript. We are issuing a decision on your manuscript at this point to prevent further delays in the evaluation of your manuscript. Please be aware that the editor who handles your revised manuscript might find it necessary to invite additional reviewers to assess this work once the revised manuscript is submitted. However, we will aim to proceed on the basis of this single review if possible. 

We look forward to receiving your revised manuscript.

Kind regards,

Miquel Vall-llosera Camps

Staff Editor

Journal Requirements:

Reviewers' comments:

Reviewer's Responses to Questions

**Comments to the Author**

1. Does this manuscript meet PLOS Global Public Health’s publication criteria ? Is the manuscript technically sound, and do the data support the conclusions? The manuscript must describe methodologically and ethically rigorous research with conclusions that are appropriately drawn based on the data presented.

Reviewer #1: Yes

2. Has the statistical analysis been performed appropriately and rigorously?

Reviewer #1: Yes

3. Have the authors made all data underlying the findings in their manuscript fully available (please refer to the Data Availability Statement at the start of the manuscript PDF file)?

Reviewer #1: Yes

4. Is the manuscript presented in an intelligible fashion and written in standard English?

Reviewer #1: Yes

5. Review Comments to the Author

Reviewer #1: I received this manuscript to review from PLOS Global Public Health and was not a reviewer for PLOS Medicine. My expertise is genetics.

The authors have already had three insightful reviews from PLOS Medicine, so I will not reiterate the general material about the paper, and will focus on responses to the reviewers’ concerns, and my additional comments.

I enjoyed reading the paper – it makes an important addition to the literature of this field and is a good fit for PLOS Global Public Health. It provides an insightful perspective on models for risk for T2D and identifies important factors for using genetics to reduce risk.

The authors have responded thoroughly and appropriately to the original reviewers’ comments, improving the completeness and the quality of the paper. I have only a few minor additional comments and suggestions.

1. A major concern from all three reviewers was the low response rate in the questionnaire – the authors have explained this carefully and performed comparative analysis where possible. This is not a major concern for me.

2. Table 2 is a valuable source of information about the study, participants and the data to be analysed. I suggest

a. Reference groups for multilevel variables, should be indicated, for easy interpretation of the beta values reported.

b. The number of missing values are added

3. In Table 2, I also lacked sufficient information to interpret the values on questionnaire items that have been added, e.g what does the mean of 11.9 (SE 1.66) mean for ‘Perceived control’ and how does this relate to the four questions in Part 6 of the questionnaire? Some information on coding to link between the questionnaire and the summary information would be useful. Apologies if I missed it.

4. The mediation effects are very important but are easier to interpret in a figure than in text. Is it possible to add an additional figure illustrating these results?

5. My one area of concern – also highlighted in limitations – was the discrepancy between the low household income (<£18k) and the high level of education attainment (college / university degree). The authors postulate reasonable explanations of immigration effects, or lack of translation of education levels in Discussion. A more worrying possibility is data quality, given these two response are the first item on each of those questions. Could the authors reassure us that there was no evidence of ‘lazy’ responding to the questionnaire, with (e.g.) the first item selected in all questions. I assume this type of QA is possible, and has been performed.

6. PLOS authors have the option to publish the peer review history of their article (what does this mean? ). If published, this will include your full peer review and any attached files.

**Do you want your identity to be public for this peer review?** For information about this choice, including consent withdrawal, please see our Privacy Policy .

Reviewer #1: No

---

## [Decision Letter · Decision Letter 1]

13 Nov 2024

PGPH-D-24-00893R1

Perceived risk of type 2 diabetes: Using linked genomic, clinical and questionnaire data to understand the potential use of genetic risk tools in British South Asians

Dear Dr. Law,

Thank you for submitting your manuscript to PLOS Global Public Health. After careful consideration, we feel that it has merit but does not fully meet PLOS Global Public Health’s publication criteria as it currently stands. Therefore, we invite you to submit a revised version of the manuscript that addresses the points raised during the review process.

The manuscript has been evaluated by two reviewers, and their comments are available below. 

Could you please carefully revise the manuscript to address all comments raised?

We look forward to receiving your revised manuscript.

Kind regards,

Steve Zimmerman, PhD

PLOS Staff Editor

Journal Requirements:

Additional Editor Comments (if provided):

Reviewers' comments:

Reviewer's Responses to Questions

**Comments to the Author**

1. If the authors have adequately addressed your comments raised in a previous round of review and you feel that this manuscript is now acceptable for publication, you may indicate that here to bypass the “Comments to the Author” section, enter your conflict of interest statement in the “Confidential to Editor” section, and submit your "Accept" recommendation.

Reviewer #1: All comments have been addressed

Reviewer #2: (No Response)

2. Does this manuscript meet PLOS Global Public Health’s publication criteria ? Is the manuscript technically sound, and do the data support the conclusions? The manuscript must describe methodologically and ethically rigorous research with conclusions that are appropriately drawn based on the data presented.

Reviewer #1: Yes

Reviewer #2: Yes

3. Has the statistical analysis been performed appropriately and rigorously?

Reviewer #1: Yes

Reviewer #2: Yes

4. Have the authors made all data underlying the findings in their manuscript fully available (please refer to the Data Availability Statement at the start of the manuscript PDF file)?

Reviewer #1: Yes

Reviewer #2: Yes

5. Is the manuscript presented in an intelligible fashion and written in standard English?

Reviewer #1: Yes

Reviewer #2: Yes

6. Review Comments to the Author

Reviewer #1: Thank you for responding to my comments - particularly appreciated the interesting information on challenges of doing QA on the questionnaire data.

Reviewer #2: Thank you for the opportunity to review “Perceived risk of type 2 diabetes: Using linked genomic, clinical and questionnaire data to understand the potential use of genetic risk tools in British South Asians.” This study is unique and makes important contributions to the literature by assessing the relative contributions of various demographic, clinical and genetic variables, as well as participants self-reported variables, to interest and primary and secondary intentions in the context of implementation of PRSs for T2D. Within the Genes & Health study, which recruited British Bangladeshi and British Pakistani volunteers, the authors conducted a questionnaire study assessing perceptions of genetic risk for T2D and primary intention to adopt health behaviors known to modify risk of developing T2D as well as secondary intention to learn further about such behaviors. The researchers also constructed a hypothesized model for structural equation modeling (SEM) analysis, conducted confirmatory factor analysis, and conducted SEM, integrating patients’ genomic and clinical data and survey results and assessing relationships between variables.

1. Abstract and introduction: The authors summarize the main research question and key findings and clearly outline the gap in the literature that the study addresses: assessing study participants’ intention to implement behavioral health changes when they are provided information about their genetic predisposition to developing T2D.

- Intro, paragraph 1, last sentence states “these findings demonstrate the unique benefit of tools such as PRSs to aid in early disease identification and management.” However, a more accurate characterization would be “to aid in identifying the degree of predisposition that individuals may have conferred by their genotypes at established T2D-associated genetic loci.”

- Intro, paragraph 2: the points made in the sentence citing reference #18 are well taken; however, the sentence is exceedingly long and it is not clear the crux of the sentence. Recommend breaking into two separate statements and going into further detail about the research cited in reference #18.

- Intro, paragraph 4, last sentence: similar to the aforementioned sentence, the sentence as written is too long. Would suggest breaking into two sentences and structuring the main objectives as “specific aims.”

- The introduction does not discuss the relative utility of PRSs versus that of “clinical and/or lifestyle-related risk factors that conventional clinical risk models rely on” for predicting the prevalence and incidence of T2D.

2. Results: The figures and table captions are complete and accurate, with appropriate presentation for the types of data included. The figures and tables also support the authors’ stated findings.

- Table 1:

- Would clarify the meaning “perceived control” and “family health behaviours” in the table itself or the figure legend.

- Would keep language regarding “primary intention” and “secondary intention” vs. “secondary interest” consistent across the manuscript and its tables/figures. Would use phrase “history of T2D” rather than “T2D history.”

- Also, did analyses account for history of gestational diabetes in family members?

- Table 2:

- Would recommend indenting the individual variables within each category of variable, or coloring the row for each category title differently for easier legibility.

- Figure 2, Figure 3, Figure 4:

- Why was “Bangladeshi ethnicity” as opposed to Pakistani ethnicity included in the hypothesized model for SEM analysis? (Figure 2)

- “first degree family with T2D history” should be changed to “first-degree relative with history of T2D” (Fig. 2 and 4).

- “Income” should be changed to “income level” to reflect the various categories of income the researchers constructed (Fig. 2 and 4).

- “including diabetes history” should be changed to to “history of diabetes” (Fig. 2 and 4).

“knowledge” should be clarified, either in the figure itself or in the figure legend (Fig. 2, 3, 4).

3. Methods

- Methods section, data analysis subsection: the hypothesised model for SEM was insufficiently described. The inter-relationships between the various variables in the hypothesised model were insufficiently justified, and the definitions of the variables (e.g., perceived control and intention) were not clearly stated in the methods section.

- A substantial amount of data were missing for T2D PRSs (19%) and BMI (35%). Further data justifying the use of multiple imputation in general, and specifically predictive mean matching, as well as possible limitations of this method, should be included in the text body.

- “T2D PRSs” and “first degree family with T2D history” were both included as variables in the SEM analysis (Fig. 2 and 4). However, potential confounding due to genetic risk that may have been conferred by, or shared with, such family members, was not discussed in this section.

4. Discussion: The authors’ discussion raises important points, citing a range of reputable articles to support their arguments. It also somewhat overreaches in the following respects:

- As discussed in section 3 above, the authors do not mention the potential confounding between study participants’ T2D PRSs and family history of T2D. The discussion also does not take into account that current health systems have limited, if any, implementation of PRSs at present (the NHS’s widespread efforts to genotype patients are exceptional compared with health systems worldwide and are not necessarily representative of other health systems). There is also insufficient discussion of the current literature comparing the relative contributions of genetic and clinical variables to predicting the incidence and prevalence of T2D.

- Limitations:

- The authors address representativeness of the study population as well as categorization of income levels as issues affecting generalizability and external validity of the study.

However, the current manuscript does not discuss limitations of the methods used in the study (see under section 3 above).

- The discussion section also does not cite publications that demonstrate that “PRSs often demonstrate the strongest clinical utility in younger populations.”

- Discussion paragraph 4:

Sentence 2, regarding the mediation of the relationship between participants’ “perceived and actual risk of T2D” overreaches. T2D PRSs do not necessarily represent “actual risk of T2D”; would use the term “T2D PRSs” instead, and throughout the manuscript, instead. Would also posit instead of stating that “this highlights the impact that previous experiences with T2D in the family context can have on how individuals think about their own risk.”

Sentence 3 is very long - would split into two.

Thank you once again for the opportunity to review this important study, which makes a valuable contribution to the field of genetic epidemiology and public health. I recommend the authors revise and resubmit their findings and am happy to review future versions or provide any clarifications on the points discussed in this review.

7. PLOS authors have the option to publish the peer review history of their article (what does this mean? ). If published, this will include your full peer review and any attached files.

**Do you want your identity to be public for this peer review?** For information about this choice, including consent withdrawal, please see our Privacy Policy .

Reviewer #1: No

Reviewer #2: No

---

## [Decision Letter · Decision Letter 2]

17 Dec 2024

PGPH-D-24-00893R2

Perceived risk of type 2 diabetes: Using linked genomic, clinical and questionnaire data to understand the potential use of genetic risk tools in British South Asians

Dear Dr. Law,

Thank you for submitting your manuscript to PLOS Global Public Health. After careful consideration, we feel that it has merit but does not fully meet PLOS Global Public Health’s publication criteria as it currently stands. Therefore, we invite you to submit a revised version of the manuscript that addresses the points raised during the review process.

The manuscript has been re-evaluated by one reviewer, and their comments are available below.

Could you please carefully revise the manuscript to address all comments raised?

We look forward to receiving your revised manuscript.

Kind regards,

Steve Zimmerman, PhD

PLOS Staff Editor

Journal Requirements:

Additional Editor Comments (if provided):

Reviewers' comments:

Reviewer's Responses to Questions

**Comments to the Author**

1. If the authors have adequately addressed your comments raised in a previous round of review and you feel that this manuscript is now acceptable for publication, you may indicate that here to bypass the “Comments to the Author” section, enter your conflict of interest statement in the “Confidential to Editor” section, and submit your "Accept" recommendation.

Reviewer #2: All comments have been addressed

2. Does this manuscript meet PLOS Global Public Health’s publication criteria ? Is the manuscript technically sound, and do the data support the conclusions? The manuscript must describe methodologically and ethically rigorous research with conclusions that are appropriately drawn based on the data presented.

Reviewer #2: Yes

3. Has the statistical analysis been performed appropriately and rigorously?

Reviewer #2: Yes

4. Have the authors made all data underlying the findings in their manuscript fully available (please refer to the Data Availability Statement at the start of the manuscript PDF file)?

Reviewer #2: Yes

5. Is the manuscript presented in an intelligible fashion and written in standard English?

Reviewer #2: Yes

6. Review Comments to the Author

Reviewer #2: Thank you once again for the opportunity to review this interesting study on an important topic, understanding the potential use of genetic risk tools in diverse populations to convey information about individuals’ risk of type 2 diabetes. Please find further comments below.

- Introduction: This section is much improved, setting the stage for discussion of the present study’s methods, results and conclusions, and I have no further comments.

- Materials and methods:

- SEM section:

- Paragraph 1: “The main exogenous variable included in our hypothesised model as shown in Fig 2 was a binary variable for age—according to the age group with the largest n in our sample—to aid interpretation (46-59 years = 1; all other age groups = 0).

- How does this binary encoding of the age variable aid in interpretation? Why was this encoding chosen, in particular because Table 2 (participant characteristics) specifies four separate age groups of various sizes?

- What are limitations of this choice?

- Paragraph 3: ”The inclusion of both T2D PRSs and family history of T2D in our model may also introduce confounding bias—however, it was in our interest to examine the potential mediating relationships between genetic and familial risk of T2D, in light of emerging evidence on their independent and complementary effects on inherited disease susceptibility.”

- Would split this into two shorter statements and repeat exact phrasing for clarity: “The inclusion of both T2D PRSs and family history of T2D in our model may introduce confounding bias. However, in light of emerging evidence of their independent and complementary effects on T2D susceptibility, we examined the potential mediating relationships between T2D PRSs and family history of T2D.”

- Discussion:

- Paragraph 5:

- “Interventions can also be supplemented with further system-level services—incorporating elements such as environmental restructuring or social planning—to facilitate the translation from risk awareness into actual preventative action across diverse groups.”

- Would break this into two separate sentences, omitting the dash, and provide examples of such systems-level services and cite examples.

- Paragraph 7:

- Very long sentence, again with unnecessary use of dash: “The inclusion of both T2D PRSs and family history of T2D in our SEM analysis may also introduce confounding bias—however, as mentioned in our methods section, it was in our interest to examine the potential mediating relationships between genetic and familial risk of T2D, in light of emerging evidence on their independent and complementary effects on inherited disease susceptibility.”

- Would break into 2 sentences and rephrase as in the SEM section: “The inclusion of both T2D PRSs and family history in our SEM analysis may have introduced confounding bias. However, as discussed in the methods section, we examined the potential mediating relationships between genetic risk of and family history of T2D.”

- Paragraph 8:

- Would clean up this sentence: “In future work, it may be that even seemingly objective sociodemographic measures also need to undergo a process of tailoring and refining to ensure that items are capturing the right categories in a diverse sample.”

- What tailoring and refining do you refer to? The previous sentences speak about representativeness of the study population; perhaps this sentence could discuss that more broadly as well.

- Paragraph 9:

- Last line is a very long sentence: “It will also be important to consider how these strategies can be tailored to be applicable to younger age groups—given the discordance found in this study between the age at which individuals would want to find out about their genetic risk of T2D, versus the age at which they would actually be ready, or willing, to implement preventative lifestyle changes.”

- Would split into two and clarify: “In the present study, we found that there was a discordance between the age group at which individuals would want to learn about their genetic risk of T2D versus the age at which they would be ready and willing to implement preventative lifestyle changes. It will be important to consider how the integration of genomics into heath systems can be tailored to younger populations.”

7. PLOS authors have the option to publish the peer review history of their article (what does this mean? ). If published, this will include your full peer review and any attached files.

**Do you want your identity to be public for this peer review?** For information about this choice, including consent withdrawal, please see our Privacy Policy .

Reviewer #2: No

---

## [Decision Letter · Decision Letter 3]

8 Jan 2025

Perceived risk of type 2 diabetes: Using linked genomic, clinical and questionnaire data to understand the potential use of genetic risk tools in British South Asians

PGPH-D-24-00893R3

Dear Miss Law,

We are pleased to inform you that your manuscript 'Perceived risk of type 2 diabetes: Using linked genomic, clinical and questionnaire data to understand the potential use of genetic risk tools in British South Asians' has been provisionally accepted for publication in PLOS Global Public Health.

Best regards,

Julia Robinson

Executive Editor

Reviewer Comments (if any, and for reference):

Reviewer's Responses to Questions

**Comments to the Author**

1. If the authors have adequately addressed your comments raised in a previous round of review and you feel that this manuscript is now acceptable for publication, you may indicate that here to bypass the “Comments to the Author” section, enter your conflict of interest statement in the “Confidential to Editor” section, and submit your "Accept" recommendation.

Reviewer #2: All comments have been addressed

2. Does this manuscript meet PLOS Global Public Health’s publication criteria ? Is the manuscript technically sound, and do the data support the conclusions? The manuscript must describe methodologically and ethically rigorous research with conclusions that are appropriately drawn based on the data presented.

Reviewer #2: Yes

3. Has the statistical analysis been performed appropriately and rigorously?

Reviewer #2: Yes

4. Have the authors made all data underlying the findings in their manuscript fully available (please refer to the Data Availability Statement at the start of the manuscript PDF file)?

Reviewer #2: Yes

5. Is the manuscript presented in an intelligible fashion and written in standard English?

Reviewer #2: Yes

6. Review Comments to the Author

Reviewer #2: All of my comments have been addressed; I have no further recommendations for the authors. Thank you for the opportunity to review your interesting manuscript.

7. PLOS authors have the option to publish the peer review history of their article (what does this mean? ). If published, this will include your full peer review and any attached files.

**Do you want your identity to be public for this peer review?** For information about this choice, including consent withdrawal, please see our Privacy Policy .

Reviewer #2: No
